# In Vitro Probiotic Properties and In Vivo Anti-Ageing Effects of *Lactoplantibacillus plantarum* PFA2018AU Strain Isolated from Carrots on *Caenorhabditis elegans*

**DOI:** 10.3390/microorganisms11041087

**Published:** 2023-04-21

**Authors:** Laura Pompa, Arianna Montanari, Alberta Tomassini, Michele Maria Bianchi, Walter Aureli, Alfredo Miccheli, Daniela Uccelletti, Emily Schifano

**Affiliations:** 1Department of Biology and Biotechnology “C. Darwin”, Sapienza University of Rome, Piazzale Aldo Moro 5, 00185 Rome, Italy; pompa.1733931@studenti.uniroma1.it (L.P.); ari.montanari@uniroma1.it (A.M.); michele.bianchi@uniroma1.it (M.M.B.); emily.schifano@uniroma1.it (E.S.); 2R&D, Aureli Mario S.S. Agricola, Via Mario Aureli 7, 67050 Ortucchio, Italy; tomassinialberta@gmail.com (A.T.); produzione@aurelimario.com (W.A.); 3Department of Environmental Biology, Sapienza University of Rome, Piazzale Aldo Moro 5, 00185 Rome, Italy; alfredo.miccheli@uniroma1.it; 4NMR-Based Metabolomics Laboratory (NMLab), Sapienza University of Rome, Piazzale Aldo Moro 5, 00185 Rome, Italy

**Keywords:** probiotic, carrots, *L. plantarum*, *C. elegans*

## Abstract

Lactic acid bacteria (LAB) share and provide several beneficial effects on human health, such as the release of bioactive metabolites, pathogen competition, and immune stimulation. The two major reservoirs of probiotic microorganisms are the human gastro-intestinal tract and fermented dairy products. However, other sources, such as plant-based foods, represent important alternatives thanks to their large distribution and nutritive value. Here, the probiotic potential of autochthonous *Lactiplantibacillus plantarum* PFA2018AU, isolated from carrots harvested in Fucino highland, Abruzzo (Italy), was investigated through in vitro and in vivo approaches. The strain was sent to the biobank of Istituto Zooprofilattico Sperimentale della Lombardia ed Emilia Romagna in Italy for the purpose of patent procedures under the Budapest Treaty. The isolate showed high survival capability under in vitro simulated gastro-intestinal conditions, antibiotic susceptibility, hydrophobicity, aggregation, and the ability to inhibit the in vitro growth of *Salmonella enterica* serovar *Typhimurium*, *Listeria monocytogenes*, *Pseudomonas aeruginosa*, and *Staphylococcus aureus* pathogens. *Caenorhabditis elegans* was used as the in vivo model in order to analyse prolongevity and anti-ageing effects. *L. plantarum* PFA2018AU significantly colonised the gut of the worms, extended their lifespan, and stimulated their innate immunity. Overall, these results showed that autochthonous LAB from vegetables, such as carrots, have functional features that can be considered novel probiotic candidates.

## 1. Introduction

In recent years, interest in the host health-promoting effects exerted by different lactic acid bacteria (LAB) strains has been increasing. As a result, several probiotic products have been developed to improve health [1]. Indeed, probiotics are known to enhance gut microbiota balance, confer protection against potential pathogenic bacteria, and prevent and/or cure intestinal diseases [2,3]. In particular, numerous bacterial strains belonging to the *Lactobacillus* genus are commonly used as probiotics and are accepted as safe by the US Food and Drug Administration and the European Food Safety Authority. Probiotics are defined as “live microorganisms which confer a health benefit on the host when administered in adequate amounts” [4]. To be considered probiotics, microorganisms must possess several features; among them, they must be able to survive through the gastro-intestinal tract, maintain viability, and potentially colonise the human host [5]. Furthermore, a micro-organism must be safe for human ingestion and should provide one or more benefits to human health, including the prevention or treatment of acute or antibiotic-associated diarrhoea, irritable bowel syndrome, inflammatory bowel disease, necrotising enterocolitis, and other potential effects, including the treatment of obesity [6,7]. These diseases are associated with an altered composition of host microbiota, with which probiotics can interact, for example, by triggering the competitive exclusion of colonisation by pathogenic bacteria. Another possible mechanism is the interaction of probiotics with the host’s immune system, leading to the modulation of the immune response [8]. The two major reservoirs of probiotic micro-organisms are the human gastro-intestinal tract and fermented dairy products. However, other sources, such as plant-based foods, represent important alternatives, since the strains isolated from these foods may be more viable and useful to formulate similar, non-dairy-based probiotic products [9]. LAB are a large heterogeneous group of Gram-positive and non-spore-forming bacteria, which have been categorised according to a set of morphological, metabolic, and physiological characteristics, mainly belonging to the *Streptococcus*, *Aerococcus*, *Pediococcus*, *Leuconostoc*, and *Lactobacillus* genera [10,11]. Among the various *Lactobacillus* strains regarded as probiotics, *Lactiplantibacillus plantarum* (formerly *Lactobacillus plantarum*) has been widely studied. Selected strains belonging to the *L. plantarum* species are extensively used as probiotics in food formulations and habitats, including fermented dairy products, sourdoughs, fruits, vegetables, cereals, meat, fish, and the mammalian gastro-intestinal tract [12]. It has been considered a probiotic because of its resistance to acid and bile, as well as its good growth characteristics that allow it to survive and persist within the gastro-intestinal tract [13]. This species has also been reported to be highly resistant to the stresses generated by technological processes and to have a great adhesion ability to the intestinal epithelial layer, thus allowing for the subsequent inhibition of the growth and adherence of several pathogens [14].

Currently, growing interest in the prolongevity effects of probiotics has led to a need for convenient in vivo models to understand the mechanisms of probiotic activity. In recent years, the nematode *Caenorhabditis elegans* has become a powerful in vivo model with which to study host–probiotic interactions. Its advantages include its ease of handling, transparency of the body, short lifespan, and absence of ethical issues. Micro-organisms represent the only food source for nematodes, which pass through the pharynx to the gut and can influence the nematode’s physiology through its metabolism [15]. Moreover, *C. elegans* is a model organism to identify new, potentially probiotic strains with specific probiotic characteristics [16,17]. This advantage is favoured by the possibility of easily monitoring anti-ageing markers, as well as body fat storage [18]. 

It has been reported that foodborne LAB-induced beneficial effects in *C. elegans*, influencing nematode longevity, larval development, fertility, lipid accumulation, and gene expression are related to fat metabolism and immunity [19,20,21].

This study aimed to determine the probiotic potential of a *L. plantarum* strain isolated from carrots. The potential probiotic properties were evaluated by in vitro tolerance to gastro-intestinal conditions, antibiotic susceptibility, aggregation features, and antagonism toward foodborne pathogenic microorganisms. In vivo tests using the *C. elegans* animal model were performed to analyse the possible beneficial effects on worms’ lifespan, gut colonisation, the ageing process, and pathogen resistance.

## 2. Materials and Methods

### 2.1. Identification of L. plantarum PFA2018AU and Growth Conditions

A lactic acid bacterial strain was isolated after carrot epidermis homogenisation, as described in [22], plating the serial dilution aliquots on De Man Rogosa Sharpe (MRS) medium, with anaerobic growth for 24–48 h at 30 °C. The carrot cultivar (*Daucus carota* L., Nantese Dordogne, Syngenta seeds) was grown in the Fucino highland (Abruzzo, Italy) and was provided by Aureli Mario S.S. Agricola (Ortucchio, AQ, Italy). Morphologically different colonies were streaked on new MRS plates and grown at 30 °C to isolate purified strains. For bacterial identification, DNA was extracted and amplified according to [23], and the 16S rDNA region of the LAB isolates was amplified using the primer pairs F8 (5′-AGAGTTTGATCCTGGCTCAG-3′) and R1492 (5′-GGTTACCTTGTTACGACTT-3′). FASTA sequences of the amplified region revealed the presence of a *Lactiplantibacillus plantarum* strain.

The *Lactiplantibacillus plantarum* PFA2018AU isolate and *L. plantarum subsp. plantarum* DSM 20174, which was used as the LAB reference strain, were grown in MRS medium at 37 °C under anaerobic conditions. For in vitro resistance to pathogens, Gram-negative *Pseudomonas aeruginosa* ATCC 15692, *Salmonella enterica* serovar Typhimurium LT2, Gram-positive *Staphylococcus aureus* ATCC 25923 and *Listeria monocytogenes* OH were used. For *C. elegans* experiments, the *Escherichia coli* OP50 strain was used as the standard food. *E. coli* OP50 and pathogen strains were grown in Luria–Bertani (LB) broth at 37 °C overnight, under shaking. 

### 2.2. Resistance to Lysozyme, Acid pH, and Bile Salts

The PFA2018AU and DSM20174 strains were grown in MRS broth overnight at 37 °C. In vitro resistance to gastro-intestinal conditions was tested according to [22]. For lysozyme resistance assays, overnight culture was centrifuged at 6000 rpm for 15 min and suspended in the same volume of SES buffer (0.22 g/L CaCl_2_, 6.2 g/L NaCl, 2.2 g/L KCl, 1.2 g/L NaHCO_3_) containing 0.1 mg/mL of lysozyme (Sigma-Aldrich, St. Louis, MO, USA). After 30 min and 2 h of incubation at 37 °C, 100 μL LAB suspensions, after serial dilutions, were seeded on MRS agar plates and further incubated at 37 °C for 24 h under anaerobic conditions. SES buffer without lysozyme was used as a control. For acid tolerance assays, 1 mL aliquot of each overnight culture (10^9^ cfu/mL) was inoculated into 9 mL of sterile phosphate-buffered saline (9 g/L NaCl, 9 g/L Na_2_HPO_4_ 2H_2_O, 1.5 g/L KH_2_PO_4_) adjusted to pH 3.0 with 8M HCl. The tubes were incubated at 37 °C for 3 h, and the viable organisms were recovered after plating on MRS agar and incubation at 37 °C. For resistance to bile salts, the same protocol was performed, using phosphate-buffered saline (PBS) with 0.3% bile salts (Sigma-Aldrich). The viability was measured as percentage CFU = [(CFU_treated_/mL)/(CFU_untreated_/mL)] × 100. The untreated value corresponds to the plate counts of inoculated bacteria in the control phosphate-buffered saline, and the treated value corresponds to the bacterial counts obtained after incubation in simulated GI conditions. Each test was repeated in triplicate.

### 2.3. Evaluation of Hydrophobicity, Auto-Aggregation and Co-Aggregation Ability

For the hydrophobicity and auto-aggregation tests, overnight cultures were centrifuged at 6000 rpm for 10 min and washed twice using PBS 1X, adjusting OD_600_ to 0.6 (A_0_). To evaluate hydrophobicity, 1 mL of xylene (apolar, aromatic solvent), toluene (monopolar, aromatic solvent), chloroform (monopolar acidic solvent), or ethyl acetate (monopolar basic solvent) was added and pre-incubated at room temperature for 10 min. The two-phase system was then vortexed for 15 s and incubated at 37 °C. The aqueous phase was removed, and the OD_600_ was determined after 2 h of incubation (A_f_). The percentage of LAB adhesion to the solvent was calculated as [(A_0_ − A_f_)/A_0_] × 100. Hydrophobicity was indicated as the percentage decrease in the optical density of the original bacterial suspension due to cells partitioning into a hydrocarbon layer.

For the auto-aggregation assay, each LAB suspension was incubated at 37 °C, and OD_600_ was measured at 1, 2, 3, 4, 5, and 24 h (A_f_). The percentage of LAB auto-aggregation was calculated as [(A_0_ − A_f_)/A_0_] × 100. For the co-aggregation test between LAB and pathogenic strains, 2 mL of the LAB and pathogen (10^8^ cells/mL) were mixed and incubated at 37 °C. The OD_600_ of the mixtures was monitored after 5 h of incubation. The absorbance of bacterial suspensions alone was also measured. The co-aggregation percentage was calculated as follows: [(A_LAB_ + A_PAT_) − 2(A_MIX_)/(A_LAB_ + A_PAT_)] × 100. Each test was performed in triplicate.

### 2.4. Antibiotic Resistance

The susceptibility test was performed according to [24]. In total, 100 μL of overnight cultures of *L. plantarum* PFA2018AU or *L. plantarum* subp. *plantarum* was plated onto MRS agar plates. Antibiotic discs were then placed on the plates, which were incubated under anaerobic conditions for 24 h at 37 °C. The zones of growth inhibition were measured from the centre of the disc, recorded, and compared with those of the DSM20174 reference strain. The test was performed in triplicate.

### 2.5. Antimicrobial Activity

The agar diffusion test was performed using (as indicator strains) *P. aeruginosa* ATCC 15692 and *S. aureus* ATCC 25923. In total, 100 μL of each LAB overnight culture was spotted onto MRS agar and coated with 5 mL of LB soft agar (0.7%); they were previously inoculated with 500 μL of each pathogen indicator strain. Plates were incubated at 37 °C for 24 h. The antagonist activity was recorded as the diameter (cm) of the growth inhibition halo around each spot. The test was repeated three times.

### 2.6. C. elegans Lifespan and Fertility Assay

A synchronous wild-type N2 *C. elegans* strain was grown at 16 °C on peptone-free Nematode Growth Medium (NGM) agar plates, plated with *L. plantarum* PFA2018AU, *L. plantarum* subp. *plantarum* DSM20174, or *E. coli* OP50. As described in [25], overnight cultures were centrifuged for 15 min at 6000 rpm. The pellet was weighed and suspended in M9 buffer (3.0 g/L KH_2_PO_4_ 6.0 g/L Na_2_HPO_4_ 0.5 g/L NaCl, 1.0 g/L NH_4_Cl) to obtain a final concentration of 400 mg/mL. Then, 25 µL of each type of bacterial lawn was plated on mNGM, and 60 worms per condition were transferred daily to new plates, plated with fresh bacterial cultures, and monitored. A worm was considered dead when it did not respond to touch. For fertility assays, synchronised worms were incubated at 16 °C on mNGM plates seeded with different bacterial strains, allowing for embryo laying. Animals were transferred onto new plates every day, and the number of progenies was counted until the mother worms became infertile. Experiments were performed in triplicate.

### 2.7. Colonisation Analysis

For each condition, 10 L4 larvae or 5-day-old adults were washed in M9 buffer and lysed, as described in [26]. Lysates were then diluted and plated onto MRS-agar plates. The number of colony-forming units (CFU) was counted after 24 h of incubation at 37 °C, anaerobically. *E. coli* OP50-fed worm lysates were plated onto LB-agar and incubated at 37 °C. The experiment was carried out in triplicate.

### 2.8. Ageing Markers Analysis

For the ageing markers, 10 worms, at the stage of 10 days of adulthood, were analysed for each condition. For the pharyngeal pumping rate, grinder contractions were analysed under a Zeiss Axiovert 25 microscope and measured within 30 s. Their locomotion capability was analysed by placing nematodes in 10 μL of M9 buffer and counting the body bending after 30 s. 

For lipofuscin accumulation, the nematodes were washed in M9 buffer and placed onto a 3% agar pad containing 20 mM of sodium azide. Then, worms were observed under a Zeiss Axiovert 25 microscope and the median fluorescence intensity (MFI) was analysed using ImageJ software 1.52 t, measuring the ratio of pixels per area of the worm. Experiments were performed in triplicate.

### 2.9. Real Time qPCR

At the stage of 1-day-old adults, 200 worms for each condition were lysed, and the total RNA was extracted, as described in [18]. *sek-1*, *skn-1*, *daf-2*, *daf-16*, *gst-4*, and *sod-3* mRNA levels were determined by quantitative real-time PCR. The differences between the mean CT values of each sample and the CT values of the housekeeping gene (*act-1*) were calculated. The primers used in this study are reported in Appendix A. The experiment was carried out in triplicate.

### 2.10. Statistical Analysis

All experiments were performed at least in triplicate. Data are presented as mean ± SD. The statistical significance was determined by Student’s *t*-test or one-way ANOVA coupled with a Bonferroni post-test (GraphPad Prism 5.0 software, GraphPad Software Inc., La Jolla, CA, USA). Differences in the *p*-values of <0.05 were considered significant and were indicated as follows: * *p* < 0.05, ** *p* < 0.01 and *** *p* < 0.001.

## 3. Results

In this study, the potential probiotic properties of a bacterial strain isolated from carrots were evaluated in vitro and in vivo. The microbial LAB population present on the carrot epidermis was isolated, as described in Section 2 Species identification at the molecular level was performed by the amplification and sequencing of 16S rDNA. The sequences obtained were compared to those in the BLAST database. A new strain of *Lactiplantibacillus plantarum* was identified and named *L. plantarum* PFA2018AU. 

### 3.1. Resistance to Lysozyme, Low pH, and Bile Salts

The probiotic properties of LAB include their ability to resist the adverse conditions of the gastro-intestinal tract (GI), managing to colonise it and provide benefits to the host. In particular, the capability of the *L. plantarum* PFA2018AU strain to survive in the presence of lysozyme, which is naturally present in saliva, and the acidic pH of the stomach and bile produced in the upper part of the intestine, was tested in vitro (Figure 1). As shown in Figure 1A, the bacteria survival rate was tested after 30 and 120 min of 1 mg/mL lysozyme treatment. After only 30 min of incubation, the *L. plantarum* PFA2018AU strain showed a survival rate that was 25% higher than the cell percentage recovered at 0 min. Moreover, the PFA2018AU strain had good tolerance at pH 3 after 2 h or 4 h of treatment, exhibiting a similar survival rate when compared to the *L. plantarum* control strain (Figure 1B). The ability of the PFA2018AU isolate to survive in the presence of 0.3% bile was tested, highlighting its resistance even after 4 h of treatment (in a similar way to the control) (Figure 1C).

### 3.2. Hydrophobicity, Aggregation, and Co-Aggregation Properties

The hydrophobicity of bacterial strains is correlated with their ability to adhere to hydrocarbons. In order to test the hydrophobicity of *L. plantarum* PFA2018AU, xylene (apolar, aromatic solvent), toluene (monopolar, aromatic solvent), chloroform (monopolar acidic solvent), or ethyl acetate (monopolar basic solvent) were used. As shown in Figure 2A, the isolated PFA2018AU strain had a high level of hydrophobicity to the solvent, similar to the probiotic control. Interestingly, *L. plantarum* PFA2018AU showed a reduction of 10% hydrophobicity using xylene but a 30% increase in hydrophobicity toward chloroform when compared to the control.

A good feature of a probiotic strain is its ability to form cellular aggregates, contributing to bacterial persistence in the intestine. As illustrated in Figure 2B, although, after the first hour of treatment, *L. plantarum* PFA2018AU showed a similar auto-aggregation with respect to the reference strain; after 4 h, an increase of about 10% was observed when compared to the control. Furthermore, probiotic strain co-aggregation with a pathogenic strain represents a good feature for protecting the intestinal tract against pathogen colonisation and preventing biofilm formation. Particularly, Figure 2C shows that the ability of *L. plantarum* PFA2018AU to aggregate with *S. aureus* was similar to that of the control with the same pathogen.

### 3.3. Resistance to Antibiotics and Anti-Pathogenicity

Probiotic bacterial strains must be susceptible to antibiotics in order to minimise their opposition to the detrimental phenomenon of the diffusion of antibiotic resistance through the bacterial population. For this reason, *L. plantarum* PFA2018AU resistance to 20 different antibiotics, including inhibitors of the synthesis of cell walls, DNA, RNA, and proteins and inhibitors of membrane function, was analysed. As shown in Table 1, *L. plantarum* PFA2018AU showed an antibiotic susceptibility pattern similar to the control strain. In contrast to the reference strain, the tested bacterial strain was also susceptible to cefuroxime, cefotaxime, and streptomycin.

Antimicrobial activity represents the ability of a probiotic to produce antimicrobial molecules, such as bacteriocins, against pathogens. For this reason, the antagonistic activity exerted by *L. plantarum* PFA2018AU against common pathogens, such as the Gram-positive *Staphylococcus aureus* and *Listeria monocytogenes*, and the Gram-negative *Pseudomonas aeruginosa* and *Salmonella enterica* serovar typhimurium LT2, was evaluated. The inhibition halo diameters produced by the isolate were comparable to that of the control against the four pathogens, as indicated in Table 2. 

### 3.4. Impact on C. elegans Lifespan and Colonisation Capability 

It is known that feeding nematodes with probiotic strains engenders prolongevity effects [27,28]. For this reason, an in vivo lifespan test was performed to analyse the potential beneficial effects exerted by the isolated strain on the *C. elegans* host. The experiment was performed by feeding the nematodes on an *L. plantarum* control, the standard feed strain *E. coli* OP50, and the *L. plantarum* PFA2018AU isolate separately, starting from when the embryos hatched. Figure 3A shows the effect of the *L. plantarum* PFA2018AU strain on the extension of the worms’ viability, with 50% viability recorded at day 28, whereas the worms fed on the *E. coli* OP50 and *L. plantarum* strains showed 50% viability on days 15 and 25, respectively. Moreover, the fertility analysis demonstrated that the brood size of the worms fed on *L. plantarum* PFA2018AU showed a reduction of about 50% and 60%, regarding the number of progeny, when compared to the probiotic reference and OP50 control, respectively (Figure 3B).

When nematodes feed on bacteria, some microbial cells escape grinder contraction and reach the intestine intact [28]. As probiotic strains must colonise the intestine to benefit the nematode, this gut colonisation capability was explored by plating lysate worms at different time points and via CFU counting. At the L4 stage, the CFU number, relative to *L. plantarum* PFA2018AU, showed results similar to those of the control. In contrast, at the stage of 5 days of adulthood, the colonisation rate was 2.0-fold higher than that of DSM and 2.5-fold higher than that of OP50 (Figure 3C). 

### 3.5. Evaluation of Ageing Processes and Innate Immunity Stimulation

In order to investigate whether the prolongevity effects exerted by the *L. plantarum* isolate correlated to a delay in ageing, age-related biomarkers, such as pumping, locomotion, and lipofuscin accumulation, were analysed in the 10-day-old adult worms. The pharyngeal pumping rate measures the grinder contractions associated with their ability to intake food, which normally declines with age. At 10 days of adulthood, the worms fed on *L. plantarum* PFA2018AU showed a significant increase in pumping rate (Figure 4A). Interestingly, the worms fed on the PFA2018AU strain showed an increase of about 40% and 20% in the number of grinder contractions when compared to OP50 and *L. plantarum* DSM 20174, respectively (Figure 4A). In the case of the locomotion analysis, a slight increase (about 20%) in body bends was observed in the 10-day-old adult worms fed on the PFA2018AU strain when compared to the *L. plantarum* reference (Figure 4B). Furthermore, the accumulation of auto-fluorescent lipofuscin was significantly reduced in those worms fed on the PFA2018AU isolate when compared to the OP50-fed nematodes, as evidenced by the median fluorescence analysis (Figure 4C,D), but this was similar to the probiotic control.

Since a good probiotic can activate the host’s immunity, the involvement of innate immunity pathways (IIS and p38 MAPK signalling) in nematodes was evaluated. The transcript levels of the *daf-2*, *daf-16*, *sek-1*, *skn-1*, *gst-4*, and *sod-3* genes involved in these cascades were analysed. A significant increase in the *daf-16*, *sek-1*, and *gst-4* transcripts was observed in the *L. plantarum* PFA2018AU-fed nematodes when compared to the controls (Figure 5). On the other hand, the expression of the *skn-1* gene was higher than in the OP50 control but was similar to *L. plantarum* DSM 20174. In summary, the general activation of genes involved in innate immunity was observed.

## 4. Discussion

The use of probiotics for human and animal health is continuously increasing. Although probiotic isolates commonly come from dairy foods, more recently, probiotic products derived from sources other than milk products are being considered. This occurs in order to diversify the use of these microorganisms by population since dairy matrices have a limited level of consumption by individuals who are, for example, allergic to milk protein, lactose intolerant, or vegan [29]. Moreover, the autochthonous probiotics isolated from alternative food matrices possess inherent stability, contributing to improvements in the survival rate in food matrices; thus, the characterisation and identification of probiotics isolated from fruits and vegetables can provide resources for manufacturing products with higher stability and production efficiency [30]. The differences in the available probiotic species or strains in food sources depend on various factors, such as the diverse features in the raw materials and the ingredients used to make unfermented or fermented foods. Indeed, probiotic microorganisms can be screened from matrices such as fruit juices, grains, honeycombs, and soil [31]. Furthermore, the native fruits of each geographical area show different properties and different associations with specific microbial communities. For example, carrot juice made from carrot roots of the same cultivar grown in three different regions in Italy showed different metabolic profiles, which were related to the different pedoclimatic conditions [32]. In this context, the main objective of this work was the characterisation of the *Lactilactibacillus plantarum* PFA2018AU strain isolated from carrots grown in the Fucino highland (Abruzzo, Italy), and the screening of its probiotic properties. The root carrot (*Daucus carota* L.) is one of the most important vegetables cultivated and consumed worldwide, and is rich in bioactive compounds, dietary fibre, antioxidants, and carotenoids [22]. Carrot-associated LAB include well-known generalist taxa, such as *Lactilplantibacillus plantarum*, *L. fermentum*, *Leuconostoc mesenteroides*, and *Weissella soli* [22,33]. 

For probiotic screening, after the initial isolation of the appropriate culture medium, the potential probiotics must meet certain requirements, including being acid- and bile-tolerant strains, not pathogenic to humans (but must possess the ability to act against pathogens in the gastro-intestinal tract), and not being able to transfer any antibiotic resistance genes to other bacteria [31,34]. Accordingly, *L. plantarum* PFA2018AU was able to resist the adverse gastro-intestinal conditions (pH 3 and bile salts) and showed good aggregation properties that were similar to the *L. plantarum* reference. Indeed, auto-aggregation or co-aggregation assays are normally performed in order to study the adhesion properties of bacterial cells, which are usually associated with the characteristics of the cell surface [35]. In humans, adhesion to epithelial cells is the first step for the subsequent colonisation of the gastro-intestinal tract, helping probiotics to compete with potential pathogens and proliferate in the gut. A preliminary test to investigate the potential adhesion capacity of probiotic bacteria to epithelial cells is the hydrophobicity metric, which is considered to represent the affinity of microorganisms to a solvent (e.g., hexane, xylene, and toluene) [36]. Indeed, bacteria with high hydrophobicity seem to show a better ability to bind to epithelial cells [37]. Notably, *L. plantarum* PFA2018AU showed a high affinity to chloroform and a percentage of hydrophobicity toward ethyl acetate and toluene that was similar to the probiotic reference. Cell adhesion is also associated with the ability to auto-aggregate, which is calculated as the number of the physical interactions between bacterial cells settling at the bottom in a static liquid suspension [38]. Accordingly, the significant co-aggregation of the PFA2018AU isolate and the *S. aureus* pathogen was obtained. When taken together, the auto-aggregation of bacteria of the same strain or the co-aggregation of bacteria of different strains, such as pathogens, can contribute to persistence in the intestine. Furthermore, by using different groups of antibiotics, such as inhibitors of cell walls, proteins, and nucleic acid synthesis, it was observed that the *L. plantarum* PFA2018AU isolate showed a antibiotic resistance profile similar to that of the *L. plantarum* DSM 20174 control. This antibiotic susceptibility, which is shared by probiotics, is a typical property that prevents the transfer of resistance genes to potential pathogens. Moreover, probiotics are able to avoid infections by foodborne pathogens through different mechanisms, such as competitive exclusion or antimicrobial molecule production [39]. Indeed, *L. plantarum* cells are reported to share various antimicrobial features, such as the production of organic acids, cyclic dipeptides, phenylacetic acid, hydrogen peroxide, low-molecular-weight compounds, protein compounds, bacteriocins, and fatty acids [40,41]. In agreement, our results highlighted the significant inhibition halos generated by the isolated strain against different pathogens. 

In parallel, the isolated strain was also analysed in the in vivo model of *C. elegans*, which represents a useful tool for screening for potential probiotic strains [16]. It feeds on bacteria as the only food source, but a substantial number of bacterial cells escape the pharynx contractions and can proceed to colonise the nematode gut [19]. Indeed, the prolongevity effects observed in PFA2018AU-fed worms were probably due to the high colonisation rate of the isolate. These data agree with the reductive effect on fertility exerted by *L. plantarum* PFA2018AU, which was similar to that of the *L. plantarum* DSM 20174 reference. Indeed, when nematodes live longer, more energy is spent trying to survive and to combat ageing processes rather than to produce progeny [18]. Moreover, the prolongevity effects observed in the survival assays were associated with the anti-ageing process, highlighted by analysing different ageing markers, such as increasing the pharyngeal pumping rate and locomotion and reducing lipofuscin accumulation. These data further demonstrate the ability of *L. plantarum* strains to prolong nematodes’ lifespan and delay ageing, as described in previous studies [42,43,44]. 

During aging, physiological decline is associated with mitochondrial dysfunction, DNA damage, protein misfolding and aggregation, and increased levels of Reactive Oxygen Species (ROS) [45,46]. The rise in ROS levels also plays a role in several disorders, such as gastrointestinal diseases [47], and the administration of probiotics can lower the risk of diseases by restoring the redox balance in the gut [48,49]. A relevant player in determining the cellular redox state is glutathione, which conjugates with electrophilic compounds through the glutathione-*S*-transferases (GSTs), a super family of Phase II detoxification enzymes [50]. In worms, the *L. plantarum* PFA2018AU diet increased the expression of *gst-4*, encoding the *C. elegans* glutathione-*S*-transferase 4. This can be ascribed to the increased transcription of *daf-16* observed in the same worms. Indeed, nuclear translocation of DAF-16 promotes the expression of target genes, such as *gst- 4* [51]. This behavior was not observed in the nematodes fed *L. plantarum* DSM20174. Moreover, several reports showed that delaying *C. elegans* aging is associated with an enhanced resistance to oxidative stress [52,53], suggesting a plausible mode of action in *L. plantarum* PFA2018AU.

Furthermore, worms do not have adaptive immunity but only innate immune defences, like human pathways [17]. Among them, the insulin/insulin-like growth factor-1 (IIS) and p38 mitogen-activated protein kinase (p38 MAPK) are more conserved in humans and nematodes, and they can be induced by probiotics [54,55]. In agreement with these works, the real-time analysis highlighted the activation of both pathways, which suggests the stimulation of immunity in *C. elegans*; this correlates with the beneficial effects observed in vivo. Consistently, a recent work reported that an increase in worms’ lifespan, triggered by a *L. plantarum* LPJBC5 strain, was linked to the activation of oxidative stress responses [43]. Moreover, previous studies have demonstrated that the probiotic effects of the bacteria preselected using *C. elegans* have also been observed in pigs, highlighting a correlation between two animal systems [56,57].

When taken together, the in vitro and in vivo tests indicate that *L. plantarum* PFA2018AU could be a potential probiotic candidate, and they also suggest that carrots could be utilised as a safe source for isolating beneficial strains. However, further validation using >in vivo trials with more complex animal or human systems should be performed to gain a deeper understanding of their potential human-health-promoting features.

## Figures and Tables

**Figure 1 microorganisms-11-01087-f001:**
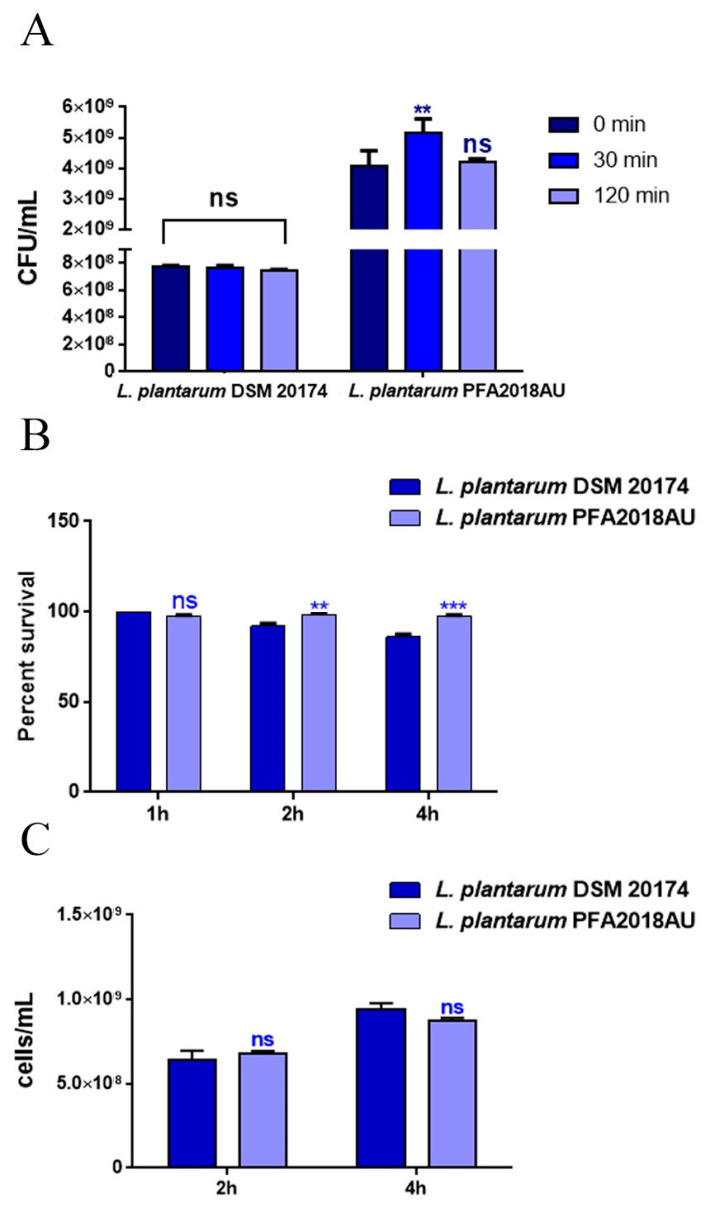
In vitro tolerance to gastro-intestinal conditions. (**A**) Resistance to lysozyme: cell counts of viable bacteria recovered at the initial time point (t0), following 30 or 120 min of incubation in 1 mg/mL of lysozyme treatment. (**B**) Recovery of viable bacteria after 2 h or 4 h of incubation in phosphate buffer adjusted to pH 3 or (**C**) 0.3% bovine bile salts. *L. plantarum* DSM 20174 was taken as a reference strain. Columns represent the mean ± SD of three independent experiments. Statistical analysis was performed via one-way ANOVA, followed by the Bonferroni post-test. Asterisks indicate significant differences (** *p* < 0.01; *** *p* < 0.001), ns: not significant.

**Figure 2 microorganisms-11-01087-f002:**
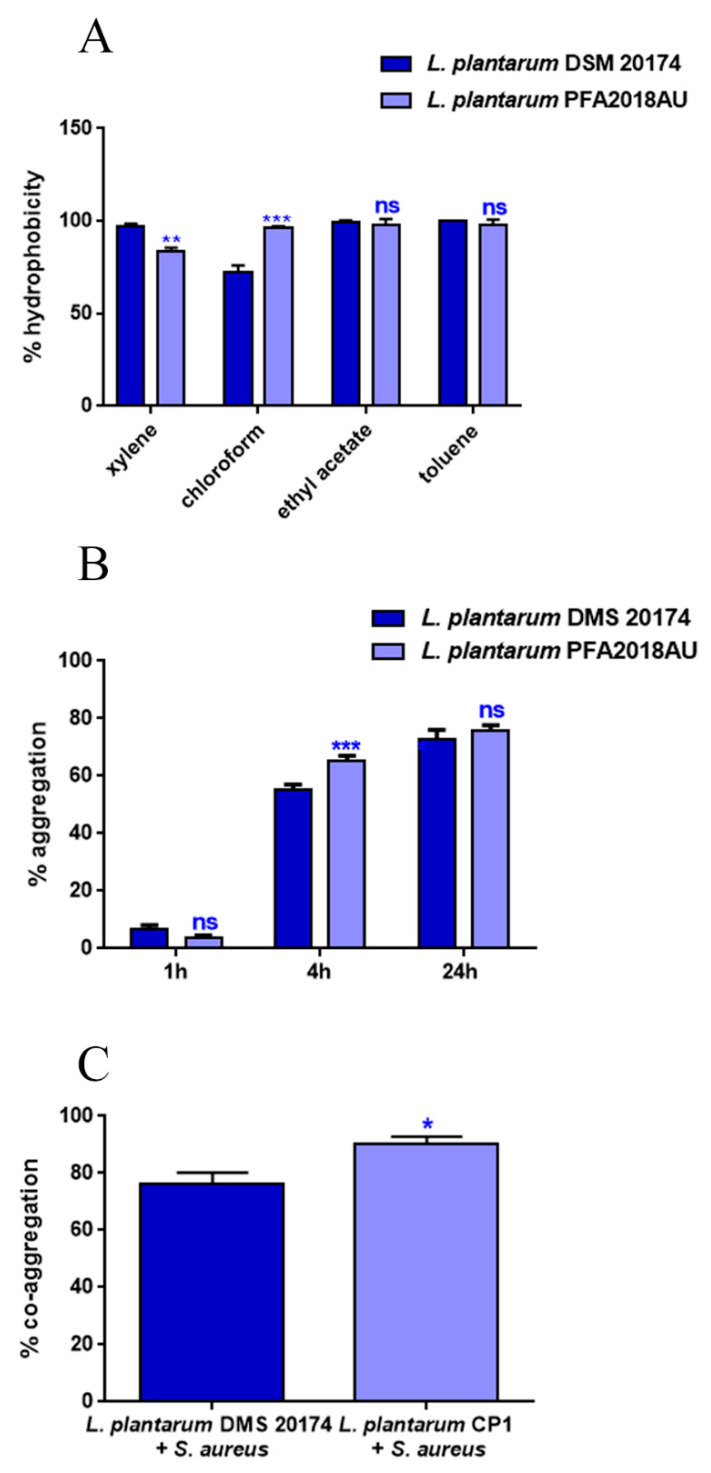
Hydrophobicity, aggregation, and co-aggregation properties of the *L. plantarum* PFA2018AU strain. (**A**) percentage of auto-aggregation measured after 24 h of incubation with different solvents and (**B**) percentage of LAB aggregation at different time points. (**C**) Percentage of co-aggregation of *L. plantarum* strains with *S. aureus* pathogen. Values are means of triplicate measurements ± standard deviation. Statistical analysis was performed via one-way ANOVA, followed by the Bonferroni post-test. Asterisks indicate significant differences (* *p* < 0.05; ** *p* < 0.01; *** *p* < 0.001), ns: not significant.

**Figure 3 microorganisms-11-01087-f003:**
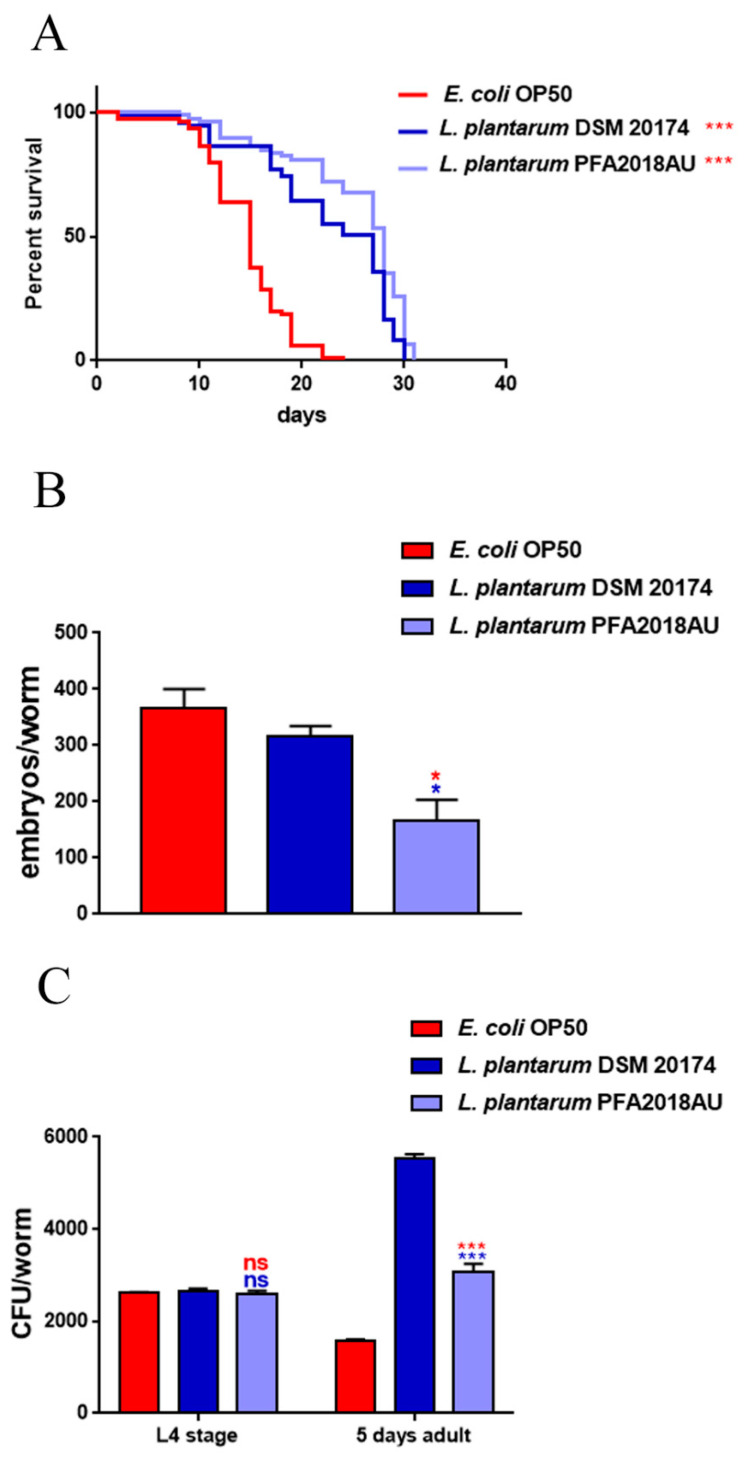
Impact of different isolates on *C. elegans* physiology. (**A**) Kaplan–Meier survival plot of N2 worms fed on the *L. plantarum* PFA2018AU strain. Lifespans of OP50- and *L. plantarum* DSM 20174-fed animals are reported as controls; n = 60 for each replicate. (**B**) Average embryo production per worm of nematodes fed with different bacterial isolates. (**C**) Bacterial colony-forming units (CFU) recovered from L4 larvae and 5-day-old adults fed with the isolate or reference strain. Bars represent the mean of three independent experiments. Asterisks indicate significant differences (* *p* < 0.05, *** *p* < 0.001) when compared to *L. plantarum* DSM 20174 (blue asterisks) or OP50 (red asterisks) controls, ns: not significant.

**Figure 4 microorganisms-11-01087-f004:**
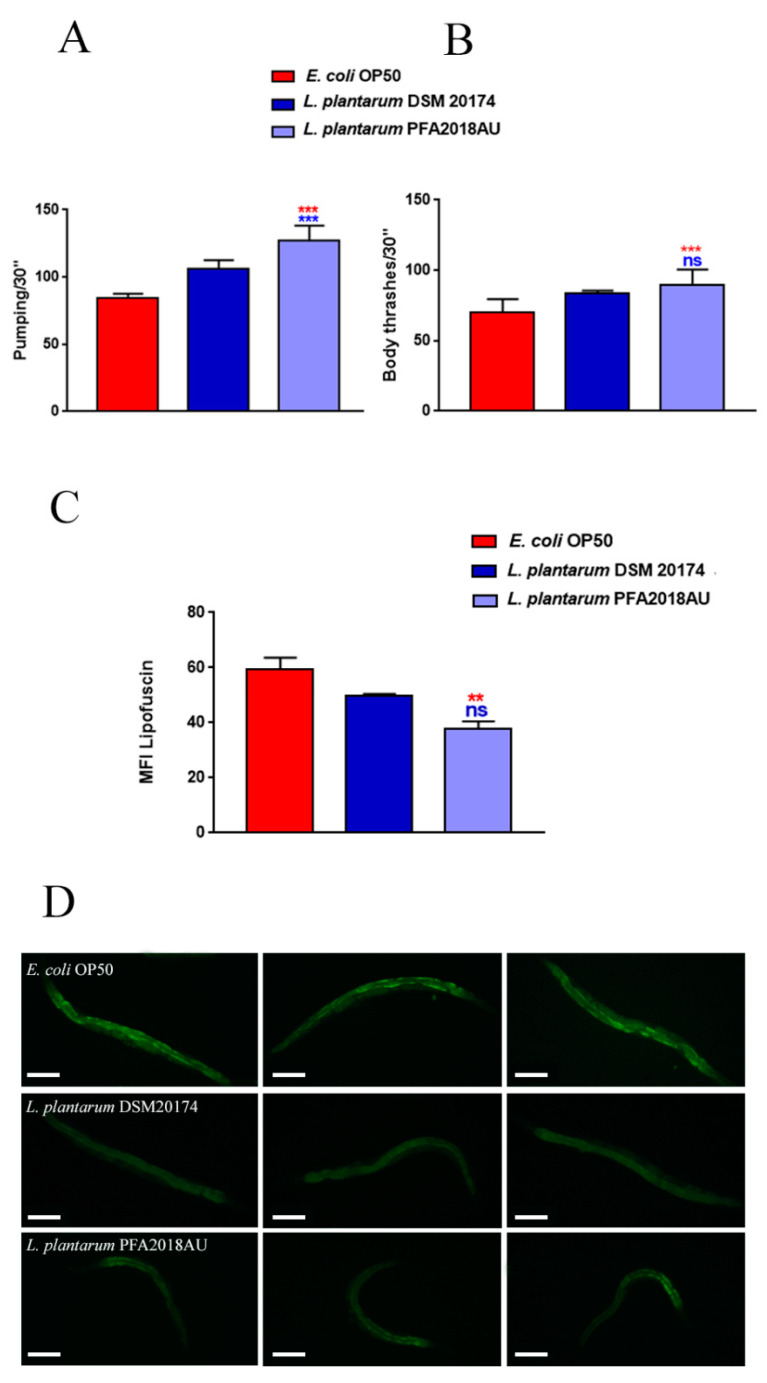
Effect of the *L. plantarum* PFA2018AU isolate on *C. elegans* ageing. (**A**) Pumping rate and (**B**) body bending of 10-day-old worms measured for 30 s. Ten worms were used for each condition. Worms fed on OP50 or *L. plantarum* DSM 20174 were used as the controls. (**C**) Mean fluorescence intensity of nematodes’ lipofuscin. Bars represent the mean of three independent experiments. Statistical analysis was evaluated via one-way ANOVA with the Bonferroni post-test. Asterisks indicate significant differences (** *p* < 0.01; *** *p* < 0.001) when compared to *L. plantarum* DSM 20174 (blue asterisks) or OP50 (red asterisks) controls, ns: not significant. (**D**) Evaluation of lipofuscin accumulation in *C. elegans* by measuring autofluorescence of lipofuscin granules fed on different LAB on day 10. Ten worms were used for each measurement. *L. plantarum* DSM 20174 (LAB reference strain) and OP50-fed worms were used as controls. Scale bar = 100 μm.

**Figure 5 microorganisms-11-01087-f005:**
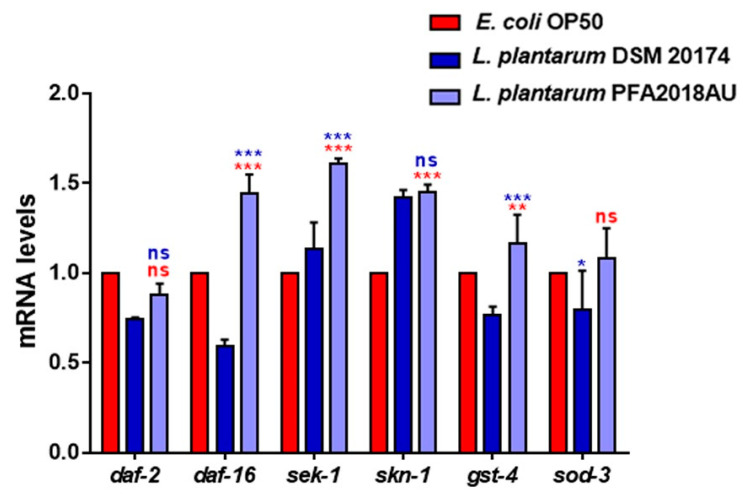
Expression of genes involved in *C. elegans* immunity. The mRNA levels of the *daf-2*, *daf-16*, *sek-1*, *skn-1*, *gst-4* and *sod-3* genes in 1-day-old adults fed on different bacterial strains. Experiments were performed in triplicate. *L. plantarum* DSM 20174 and OP50 were used as the controls. Data are presented as mean ± SD. Asterisks indicate significant differences (* *p* < 0.05, ** *p* < 0.01, *** *p* < 0.001); ns: not significant.

**Table 1 microorganisms-11-01087-t001:** Different isolates’ resistance to antibiotics. The zones of inhibition were measured from the centre of the disc and recorded in cm ± SD. Absence of inhibition halo was indicated as (0), ns: not significant.

Antibiotic	*L. plantarum* DSM 20174	*L. plantarum* PFA2018AU	*p*-Value
Vancomycin	0	0	ns
Clindamycin	0.9 cm ± 0.35	0.5 cm ± 0.20	*p* < 0.001
Cefalotin	0.1 cm ± 0.12	0.5 cm ± 0.53	*p* < 0.001
Cefuroxime	1 cm ± 0.23	0	*p* < 0.001
Tobramycin	0	0.6 cm ± 0.05	*p* < 0.001
Ampicillin	0.5 cm ± 0.20	0.8 cm ± 0.08	*p* < 0.001
Cefotaxime	0.6 cm ± 0.81	0	*p* < 0.001
Chloramphenicol	1 cm ± 0.50	0.9 cm ± 0.83	*p* < 0.05
Tetracycline	0.6 cm ± 0.24	0.8 cm ± 0.23	*p* < 0.01
Erythromycin	0.7 cm ± 0.20	0.6 cm ± 0.15	*p* < 0.05
Amikacin	0	0	ns
Oxacillin	1 cm ± 1.10	1 cm ± 1.2	ns
Fosfomycin	0	0	ns
Rifampicin	1 cm ± 0.08	0.9 cm ± 0.10	*p* < 0.05
Gentamicin	0	0	ns
Penicillin	1 cm ± 0.72	0.6 cm ± 0.31	*p* < 0.001
Aztreonam	1 cm ± 0.60	0.1 cm ± 0.50	*p* < 0.001
Carbenicillin	0.3 cm ± 0.58	1 cm ± 0.92	*p* < 0.001
Mezlocillin	1 cm ± 0.3	0.9 cm ± 0.45	*p* < 0.05
Streptomycin	0.5 cm ± 0.9	0	*p* < 0.001

**Table 2 microorganisms-11-01087-t002:** Antagonistic activity in vitro. The diameter of the inhibition halos was recorded in cm, and the data are expressed as media ± SD.

	*S. enterica*	*L. monocytogenes*	*S. aureus*	*P. aeruginosa*	*p*-Value
*L. plantarum* DSM 20174	2.2 ± 0.36	2.3 ± 0.32	2.2 ± 0.34	2.3 ± 0.5	ns
*L. plantarum* PFA2018AU	2.4 ± 0.67	2.5 ± 0.46	2.5 ± 0.45	2.7 ± 0.6	ns

## Data Availability

Not applicable.

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
