# Peer review of "In Vitro Probiotic Properties and In Vivo Anti-Ageing Effects of Lactoplantibacillus plantarum PFA2018AU Strain Isolated from Carrots on Caenorhabditis elegans"

_microorganisms, 2023, doi:10.3390/microorganisms11041087_

Round 1

Reviewer 1 Report

The manuscript is interesting and novel, but some aspects should be considered in order to improve the study. 

1. In materials and methods, it is not very clear why they use two temperatures, 30 and then 37, what was the optimal temperature for the growth of the bacteria isolated from the carrot epidermis?

2. In line 110, why is the Escherichia coli OP50 strain used as standard food, it is not clear, is a reference needed?

3. On line 124, why is pH 5 considered as control, this is not clear, since in the stomach the conditions to evaluate bacteria with probiotic potential should be pH 2 and/or pH 3.

4. What is M9 buffer?

5. On line 240, why is affinity in parentheses?

6. In Figure 2C the Y-axis label is missing.

7. In Table 1, cefuroxime has no standard deviation. It is recommended to remove the minus and plus, since they cofound, and place 0 as an indication that there is no inhibition halo.

8. The discussion is super short, it lacks much emphasis regarding the in vitro tests on probiotic potential, which makes this lactic acid bacteria isolated from carrot epidermis so special, in comparison with other commercial probiotics.

9. The discussion does not expand more on why the in vivo model used is so important, nothing is discussed about the results obtained.

10. It is not clear what the conclusion of the work is. It talks in line 438 about effects on the intestinal barrier, what did the authors measure to demonstrate this?

The authors should improve the writing of the manuscript, as well as improve the discussion of the results.

Reviewer 2 Report

The article by L. Pompa et al is devoted to the study of the properties of a potentially probiotic strain isolated from carrots and its identification. The authors conducted a series of in vitro tests and tested the effect of this strain on nematodes. In my opinion, the work presents a fairly wide range of properties of this new strain, which allow us to evaluate it as a potential probiotic.

There are a minor remark.

Line 167. Abbreviation "NGM" should include full name at first mention

In my opinion, a qualitative study has been carried out and the manuscript can be accepted for publication.

Author Response

Line 167. Abbreviation "NGM" should include full name at first mention

We thank the Reviewer for the suggestion. The text has been modified accordingly.

Round 2

Reviewer 1 Report

The authors should improve the discussion. 

In line 170 a parenthesis is missing

In table 1 the authors make the requested changes, but in the last row they did not remove the "+" sign or what does this sign mean?

The authors have many results, I still consider that they need to discuss them more, to highlight the importance of their findings, to make their work more interesting.

Author Response

The authors should improve the discussion. 
We thank the Reviewer for the comment. The discussion has been improved, as suggested.

In line 170 a parenthesis is missing
We thank the Reviewer for the suggestion. A parenthesis has been added.

In table 1 the authors make the requested changes, but in the last row they did not remove the "+" sign or
what does this sign mean?
We apologyse for the typo. We removed the “+” and a “zero” has been added.